# Key processes required for the different stages of fungal carnivory by a nematode-trapping fungus

Hung-Che Lin[1◉], Guillermo Vidal-Diez de Ulzurrun[1◉], Sheng-An Chen[1¤a], Ching-Ting Yang[1], Rebecca J. Tay[1], Tomoyo Iizuka[1], Tsung-Yu Huang[1,2], Chih-Yen Kuo[1,2], A. Pedro Gonçalves[1¤b], Siou-Ying Lin[1], Yu-Chu Chang[3], Jason E. Stajich[4], Erich M. Schwarz[5], Yen-Ping Hsueh [1,2]*

1 Institute of Molecular Biology, Academia Sinica, Nankang, Taipei, Taiwan, 2 Molecular and Cell Biology, Taiwan International Graduate Program, Academia Sinica and Graduate Institute of Life Sciences, National Defense Medical Center, Taipei, Taiwan, 3 Department of Biochemistry and Molecular Cell Biology, School of Medicine, Taipei Medical University, Taipei, Taiwan, 4 Department of Microbiology and Plant Pathology, University of California, Riverside, California, United States of America, 5 Department of Molecular Biology and Genetics, Cornell University, Ithaca, New York, United States of America

◉ These authors contributed equally to this work.
¤a Current address: Department of Biology, Duke University, Durham, North Carolina, United States of America
¤b Current address: College of Medicine, National Cheng Kung University, East District, Tainan, Taiwan
* pinghsueh@gate.sinica.edu.tw

**Data Availability Statement:** Data are available from the NCBI GEO database (http://www.ncbi.nlm.nih.gov/geo/; series record: GSE233568).

## Abstract

Nutritional deprivation triggers a switch from a saprotrophic to predatory lifestyle in soil-dwelling nematode-trapping fungi (NTF). In particular, the NTF *Arthrobotrys oligospora* secretes food and sex cues to lure nematodes to its mycelium and is triggered to develop specialized trapping devices. Captured nematodes are then invaded and digested by the fungus, thus serving as a food source. In this study, we examined the transcriptomic response of *A. oligospora* across the stages of sensing, trap development, and digestion upon exposure to the model nematode *Caenorhabditis elegans*. *A. oligospora* enacts a dynamic transcriptomic response, especially of protein secretion–related genes, in the presence of prey. Two-thirds of the predicted secretome of *A. oligospora* was up-regulated in the presence of *C. elegans* at all time points examined, and among these secreted proteins, 38.5% are predicted to be effector proteins. Furthermore, functional studies disrupting the t-SNARE protein Sso2 resulted in impaired ability to capture nematodes. Additionally, genes of the DUF3129 family, which are expanded in the genomes of several NTF, were highly up-regulated upon nematode exposure. We observed the accumulation of highly expressed DUF3129 proteins in trap cells, leading us to name members of this gene family as Trap Enriched Proteins (TEPs). Gene deletion of the most highly expressed TEP gene, *TEP1*, impairs the function of traps and prevents the fungus from capturing prey efficiently. In late stages of predation, we observed up-regulation of a variety of proteases, including metalloproteases. Following penetration of nematodes, these metalloproteases facilitate hyphal growth required for colonization of prey. These findings provide insights into the biology of

**Funding:** Funding for this work was provided by the Academia Sinica Investigator Award AS-IA-111-L02 and the Ministry of Science and Technology MOST grant 110-2311-B-001-047-MY3 to Y.-P.H. Computing was also supported by a research allocation from NSF XSEDE (TG-MCB190010) to E.M.S. The funders had no role in study design, data collection and analysis, decision to publish, or preparation of the manuscript.

**Competing interests:** The authors have declared that no competing interests exist.

**Abbreviations:** AMP, antimicrobial peptide; ECM, extracellular matrix; GO, Gene Ontology; HCR, hybridization chain reaction; hpe, hours post-exposure; HU, hydroxyurea; IDR, intrinsically disordered region; LLPS, liquid–liquid phase separation; LNM, low-nutrient medium; LPMO, lytic polysaccharide monooxygenase; MAPK, mitogen-activated protein kinase; NTF, nematode-trapping fungi; PDA, potato dextrose agar; PDB, Potato Dextrose Broth; PIC, protease inhibitor cocktail; TEP, Trap Enriched Protein.

the predatory lifestyle switch in a carnivorous fungus and provide frameworks for other fungal–nematode predator–prey systems.

## Introduction

Predator–prey relationships are a universal feature of the natural world, shaping the evolution and diversity of species throughout the tree of life. As every organism within the animal kingdom acts as either a consumer or the consumed, predator–prey relationships play a critical role in maintaining the balance of ecosystems. While primarily acting as prey, even some species of plants have evolved to capture and consume small animals, allowing them to thrive in nutrient-poor environments [1]. Similarly, in the fungal kingdom, carnivorous fungi have evolved strategies for killing and consuming nematodes, which are the most abundant animals on the planet and populate all soil ecosystems. Carnivorous fungi can be found in divergent lineages, indicating that predation in fungi has independently evolved multiple times during evolution [2–4].

In the phylum of Ascomycota, the majority of carnivorous fungi form a monophyletic group belonging to the family Orbiliaceae, and they prey on nematodes through the development of specialized trap structures [5]. Ubiquitous in the soil, these nematode-trapping fungi (NTF) usually behave as saprotrophs. However, nutritional starvation and the presence of nematodes trigger a switch towards predation, resulting in the formation of traps. Different trapping devices, including adhesive nets, columns, knobs, nonconstricting and constricting rings, are formed depending on the fungal species [6,7]. Previous work has shown that NTF behave as generalist predators and display marked natural diversity in trap formation [8]. Among fungal traps, adhesive nets are the most effective in capturing nematodes [6,7] and are most well studied in the nematode-trapping fungus *Arthrobotrys oligospora*.

Work in *A. oligospora* has enhanced our understanding of the distinct stages of NTF predation, including sensing, trap development, and digestion. Firstly, prey sensing occurs partially via the recognition of nematode ascarosides, small molecules composed of dideoxysugar ascarylose linked to a fatty acid–like side chain normally employed by nematodes for intraspecies communication [9]. Unlike higher complexity predators, NTF are sessile and unable to chase their prey. Therefore, *A. oligospora* relies on molecular mimicry of food and sex cues to attract nematodes to their vicinity. In particular, *A. oligospora* produces volatiles, such as methyl 3-methyl-2-butenoate that mimics a sex pheromone cue and is highly attractive to hermaphrodites and female nematodes [10].

After sensing prey, trap development begins. Recently, a number of *A. oligospora* genes contributing to trap formation have been identified [8]. These proteins function in adhesion [11], autophagy [12], signaling [13–22], pH sensing [23], calcium uptake [24], cytoskeletal structure [25], Woronin body assembly [26], metabolism [27,28], transcriptional regulation [16,29–31], and small RNA processing [32]. Additionally, in the related species *A. flagrans*, cell fusion controlled by SofT and the STRIPAK component SipC is required for trap closure [33–35]. While these studies have expanded our understanding of the predator–prey system formed by NTF and nematodes, the molecular mechanisms underlying *A. oligospora* predation remain largely unclear.

To gain molecular insights into the predatory lifestyle switch in NTF, we used time-course transcriptional profiling to examine global gene expression patterns in *A. oligospora* exposed to *Caenorhabditis elegans*. The resulting datasets pointed to numerous biological processes

fine-tuned by *A. oligospora* during each stage of predation. These major biological processes include up-regulation of ribosome biogenesis and DNA replication upon sensing nematode signals, enhancement of the secretory pathway during trap morphogenesis, and up-regulation of protease and transporter gene families during digestion of prey. By combining RNA-sequencing with functional analysis of candidate genes, our study sheds light on the complex molecular and cellular mechanisms involved in NTF predation on nematodes.

## Materials and methods

### Strains and culture conditions

The *A. oligospora* strains used in this study are listed in Table 1. *A. oligospora* strains were maintained on potato dextrose agar (PDA; Difco). Trap induction, capture rate, and survival rate assays were all performed on low-nutrient medium (LNM: 2% agar, 1.66 mM MgSO$_4$, 5.4 mM ZnSO$_4$, 2.6 mM MnSO$_4$, 18.5 mM FeCl$_3$, 13.4 mM KCl, 0.34 mM biotin, and 0.75 mM thiamin). *C. elegans* N2 strain was maintained on NGM with *E. coli* (OP50) as the food source.

### RNA-sequencing analyses

*A. oligospora* TWF154, grown on LNM, was transferred to fresh 9 cm LNM plates. After approximately 5 days (25°C, dark), the mycelium was exposed to approximately 2,500 to 3,000 *C. elegans* N2 L4-stage larvae per plate, and hyphal collections were taken at 2, 4, 10, 24, and 48 h after exposure, respectively. For the 2 and 4 h time points, nematodes were washed out from the Petri dishes prior to hyphal collection. The hyphae were harvested using a cell scraper and immediately frozen in liquid nitrogen. Total RNA was isolated by the Trizol-Phenol-Chloroform method, and cDNA libraries were prepared using the Illumina TruSeq Stranded mRNA kit. Prepared cDNA was sequenced on an Illumina NextSeq500 for 76 cycles (high-output), and 2 × 75 bp paired-end reads were obtained. Each biological replicate consisted of 45 to 76 M reads (approximately 85 to 144× coverage). Raw reads were filtered with fastp 0.14.1 and a minimum length of 68 was imposed, resulting in a minimum mean quality for 96% of Q30 read bases and 100% of >Q34 read bases. Salmon 0.9.1 was used to quantify transcript expression. Bootstrapped abundance estimates (200 bootstrap samples) were computed [36] and pseudoaligned using Kallisto [37]. Sleuth 0.29.0 [38] was employed to determine differential

**Table 1. *Arthrobotrys oligospora* strains used in this study.**

| Name | Strain name | Genotype |
|---|---|---|
| WT | TWF154 | |
| *ku70* WT | TWF1697 | *Δku70::hph* |
| H2B::mCherry | TWF2804 | *H2B::mCherry::hph* |
| *sso2* | TWF3532 | *Δku70::hph; ΔEYR41_000055::nat1* |
| TEP1-GFP | TWF3741 | *TEP1::GFP::Tgluc* |
| *tep1* | TWF3571 | *Δku70::hph; ΔEYR41_003105::nat1* |
| *tep1 TEP1* | TWF3585 | *Δku70::hph; ΔEYR41_003105::nat1; EYR41_003105::g418* |
| *tep3* | TWF3513 | *Δku70::hph; ΔEYR41_011738::nat1* |
| *tep4* | TWF3438 | *Δku70::hph; ΔEYR41_009694::nat1* |
| *mcp1* | TWF3522 | *Δku70::hph; ΔEYR41_010712::nat1* |
| *amp1* | TWF3677 | *Δku70::hph; ΔEYR41_005433::nat1* |
| cytosolic GFP WT | TWF3407 | *Polic::GFP::Tgluc* |
| cytosolic GFP *mcp1* | TWF3408 | *Δku70::hph; ΔEYR41_010712::nat1; Polic::GFP::Tgluc* |
| cytosolic GFP *amp1* | TWF3409 | *Δku70::hph; ΔEYR41_005433::nat1; Polic::GFP::Tgluc* |

expression using the likelihood ratio test across all conditions, resulting in $p$-values and $q$-values (adjusted $p$-values) used in all subsequent analysis. In addition, a Wald test was performed to calculate the coefficient of the full model for the studied condition, denoted by *beta*. *beta* coefficients were computed to be comparable (but not equal) to the $\log_2$ fold change between samples exposed to nematodes and those harvested before the addition of nematodes [38]. The resulting dataset is available from the NCBI GEO database (http://www.ncbi.nlm.nih.gov/geo/; series record: GSE233568). A $q$-value of $<0.01$ and a *beta* $>1$ or $<-1$ were used to define transcripts with altered expression. Sleuth [38] and Jvenn [39] were used to perform principal component analysis and build Venn diagrams, respectively. Transcript clusters were obtained for the differentially expressed genes using clust [40]. Selected lists of differentially expressed transcripts were run through Omicsbox 2.0.36 [41] for functional enrichment analysis, with $p$-value $< 0.01$ (Fisher's exact test) as the threshold and the "Remove double IDs" option. The initial results obtained for both "Interpro IDs" and "Interpro GO IDs" were consolidated using the "reduce to most specific" option ($p$-value $< 0.01$). Heat maps were built on Graph-Pad Prism.

## Secretome of *A. oligospora*

The secretome of *A. oligospora* was predicted using a combination of tools [42]. First, the entire transcriptome was run on SignalP 5.0 (cutoff = 0.8) [43], Phobius [44], and TMHMM 2.0 [45] to predict the presence of N-terminal signal peptides and transmembrane regions. Transcripts with a signal peptide (as defined by the concordant detection by SignalP and Phobius) and no transmembrane regions (according to both Phobius and TMHMM) were recorded as "list A." To enhance our confidence in the final list of predicted secreted transcripts, which may represent proteins bearing either noncanonical secretion domains or signal peptide domains mistakenly categorized as N-terminal transmembrane domains by previous programs, we employed the following methodology: Transcripts lacking a signal peptide, as determined by the absence of SignalP and/or Phobius outputs, and featuring precisely one transmembrane region exceeding 18 amino acids within the initial 60 amino acids of the encoded protein were categorized as "list B." Transcripts that exhibited a signal peptide, consistently detected by both SignalP and Phobius, and harbored one or more transmembrane domains (defined by Phobius and TMHMM, and not present in "list B") were designated as "list C." Subsequently, "list B" and "list C" were analyzed using WolfPSort [46], with transcripts achieving an "Extracellular" score of $\geq 17$ recorded as "list B1" and "list C1." Finally, list A, list B1, and list C1 were separately run on MitoFates [47]. Transcripts predicted to be targeted to mitochondria (cutoff = 0.5) were excluded, yielding the final compilation of predicted secreted transcripts. This comprehensive approach identified 913 secreted transcripts from list A (84.2%), 17 from list B1 (1.6%), and 154 from list C1 (14.2%). The final list (1,084 secreted transcripts) was run on EffectorP-fungi 3.0 [48] to detect putative effectors, yielding 417 putative effectors.

## DUF3129 gene detection and orthologs distribution

The DUF3129 domain is defined as the conserved domain with accession numbers IPR021476 (InterPro) and PF11327 (Pfam). Multiple amino acid sequence alignments were executed on MAFFT [49] and subsequently visualized using Jalview [50]. Genes in the family of DUF3129 were characterized as those containing the DUF3129 domain of TEP1. Therefore, DUF3129 genes were found using the domain as query for a BlastP against filtered model proteins of each genome. On the mycocosm database, we used the blast section of the mycocosm portal [51] with options: aligment program blastp protein versus protein, database Gene Catalog proteins, Expect 1.0E-5, Word Size 3, Filter low complexity regions, Perform gapped alignment,

and Scoring Matrix BLOSUM62. Local alignments were produced using blastp option of blast 2.9.0 [52], with options -outfmt 6 -max_target_seqs 500 -evalue 1e-5 -num_threads 16, on databases produced using makeblastdb -dbtype prot for each genome. Orthology groups for the 19 selected high-quality genomes (S5 Table) were computed using funannotate_compare within funannotate 1.8 [53], relying on ProteinOrtho v5 [54] with options -synteny -cpus = 24 -singles -selfblast. Orthogroups were selected among all the computed orthogroups as those containing at least one of the DUF3129 genes obtained from the blastp local alignment. To study gene gain and loss of the DUF3129 family, we inferred gene duplication and loss events from gene trees constructed with all DUF3129 genes in 19 high-quality fungal genomes and the species tree depicted in S3 Fig. Gene trees were constructed using the sequence alignment tool Muscle5 [55] and the software for phylogenetic tree inference, IQ-TREE 2 [56]. The species tree was built based on concatenation of single-copy orthologs from the 19 fungal genomes using Muscle5 and IQ-TREE 2. Further refinement of the gain/loss pattern was computed through SpeciesRax [57] with default settings. DUF3129 genes, obtained through blastp, that were not clustered in orthogroups of at least 4 genes were not suitable to compute gene trees; consequently, for such genes, we assessed shared lineage manually. Finally, genes without a shared lineage were counted as species-specific gained genes. The final results were summarized and visualized using TreeViewer [58]. Chromosomal maps of the DUF3129 genes were drawn using MapChart [59].

## Targeted gene deletions and complementation

Targeted gene deletions were generated from either wild-type or the *ku70* mutant of *A. oligospora* TWF154, as described previously [18]. Briefly, an overlap PCR-based construct was obtained by fusing the sequences 1.5 kb upstream and downstream of the target genes to a hygromycin resistance cassette (amplified from vector pAN7-1) or a clonNAT resistance cassette (amplified from vector pRS41N). The construct was then introduced into the protoplast of either *A. oligospora* TWF154 or the *ku70* background strain via PEG-mediated transformation. Protoplasts of *A. oligospora* were generated according to previously described methods [18]. Approximately $5 \times 10^6$ conidia were cultured in 100 mL of PDB for 24 h at 25°C and 200 rpm. Fungal hyphae were collected by centrifugation and washed by MN buffer (0.3 M MgSO$_4$, 0.3 M NaCl). The mycelium was then mixed with 5 mL of 100 mg/mL VinoTastePro enzyme mix in MN buffer overnight at 25°C and 200 rpm. Protoplasts were filtered through 3 layers of miracloth and washed with STC buffer (1.2 M sorbitol, 50 mM CaCl$_2$, 10 mM Tris–HCl (pH 7.5)). Next, $5 \times 10^5$ protoplasts were mixed with 5 μg of construct DNA and incubated on ice for 30 min. Five times volume of PTC buffer (40% PEG 4000 [w/v], 10 mM Tris–HCl (pH 7.5), 50 mM CaCl$_2$) was then added and allowed to incubate for 20 min at room temperature. Then, molten regeneration agar (1% agar, 0.5 M sucrose, 0.3% yeast extract [w/v], 0.3% casein acid hydrolysate [w/v]) containing 100 μg/mL hygromycin B or 250 μg/mL clonNAT was added to the protoplasts and poured into Petri dishes. Transformants were then screened for gene deletion. Primers used for gene deletion and genotyping are listed in S1 Table.

For gene complementation, a wild-type copy of the target gene along with flanking sequence (2 kb upstream and 1 kb downstream) was amplified, fused with a G418 marker, and transformed into mutant protoplasts. The complemented strains were then genotyped to confirm gene rescue.

## Phenotypic characterization

Induction of *A. oligospora* trap morphogenesis by *C. elegans* was performed as previously described [8]. Briefly, fungal strains were grown on 3-cm LNM plates for 48 h before exposure

to 30 *C. elegans* L4 larvae. After 6 h, nematodes were removed, and 24 h later, random images within 0.5 cm of the plate edge were captured using a Zeiss Stemi 305 microscope with a Zeiss Axiocam ERc 5s camera at 40X magnification. Trap number was quantified from the images using Fungal Feature Tracker [60].

For rapamycin treatment, the final concentration of LNM with 4 μg/mL rapamycin was prepared by directly adding 10 mg/mL rapamycin stock to LNM. To quantify the *C. elegans* survival rate, WT *A. oligospora* was inoculated onto 3-cm LNM with and without rapamycin and cultured for 3 days, 25°C. After addition of 30 L4 *C. elegans*, the number of free-living worms was counted every 3 h for a total of 12 h. To quantify nuclei number between hyphal and trap cells, trap induction +/− rapamycin was performed as above using the H2B::mCherry strain. After 24 h, fungal cultures were stained with SCRI Renaissance 2200 (SR2200), which stains fungal cell walls without arresting growth, and imaged using an LSM980 Airyscan 2 confocal microscope (Carl Zeiss). Nuclei number was manually counted within each fungal cell.

To investigate the role of DNA replication in trap morphogenesis, fungal strains grown on 3-cm LNM were incubated with the DNA replication inhibitor, hydroxyurea (HU), at a final concentration of 200 mM for 30 min before exposure to *C. elegans*.

To estimate the *C. elegans* escape rate from *A. oligospora* traps, approximately 200 L4 to young adult worms were introduced per fungal plate to induce trap formation. After 24 h, another 50 young adult worms were added to the traps for 10 min before shaking slightly and rinsing with 1 mL of deionized water. The number of uncaptured nematodes was manually counted, and the nematode escape rate was calculated as (uncaptured nematodes/50) * 100%.

To characterize the growth of fungal mycelia inside nematodes after penetration, fungal strains were transformed with a cytosolic GFP plasmid. A chunk of agar with mycelia from the mother culture was inoculated onto a 5-cm LNM plate and allowed to grow for 3 days. Following 24 h of trap induction by addition of worms, additional adult worms were inoculated onto the formed traps and imaged using the Andor Revolution WD system with a Nikon Ti-E automatic microscope and an XON Ultra 888 EMCCD camera (Andor).

## Third-generation HCR in situ hybridization

*In situ* hybridization was performed with slight modifications to the previously described third-generation *in situ* HCR method [61]. First, wild-type *A. oligospora* were grown on a 24-well LNM plate (each well approximately 1.9 cm$^2$) for 48 h. Approximately 100 young adult *C. elegans* were then added onto the fungal hyphae. After 18 h of contact, the samples were fixed with 1 mL of 1% formaldehyde in PBS for 10 min at room temperature and 50 rpm. Immediately after fixation, lysing enzyme (Sigma-Aldrich) was added to a final concentration of 100 mg/mL and incubated at room temperature for 2 h. After digestion, the lysed hyphae were permeabilized with 70% ethanol overnight at 4°C and 50 rpm, followed by Triton treatment (0.1% in PBS) for 5 min at room temperature and 50 rpm. The plate was then prewarmed to 37°C for the following steps. During detection, 200 μL of probe hybridization buffer was added to the sample for 10 min. The sample was then hybridized with a probe solution (0.4 pmol in 100 μL of probe hybridization buffer) overnight in a humidified dark chamber at 37°C. Amplification was performed overnight to generate long HCR amplification polymers, which were then imaged using an LSM980 Airyscan 2 confocal microscope (Carl Zeiss) with a C-Apochromat 63×/1.2 numerical aperture (NA) water immersion objective. Images were acquired with Multiplex Mode (SR-4Y).

## TEP1-GFP fusion protein construction

To construct the TEP1-GFP fusion protein using In-Fusion cloning, TEP1 and 2 kb upstream were amplified using the primers 3105_LGFP_F1 (5′-AGGGAACAAAAGCTGGTACCTACA

TGCAGAGTCTGTGGTTAC-3′) and 3105_LGFP_R1 (5′-ACCGCCACCGGCGCGCCTGT TACCCTTTTTCTCAGCTGGC-3′). PCR amplification was performed with Phusion High-Fidelity DNA Polymerases (NEB). The amplicon was then cloned into a GFP plasmid with a geneticin resistance cassette using the In-Fusion HD PCR Cloning Kit (Clontech).

### Intrinsically disordered regions (IDRs) and three-dimensional structure prediction

To identify IDRs in TEPs, we utilized 2 commonly used disorder predictors, that is, PONDR VSL2 [62] and PONDR VL3 [63]. These predictors can be accessed at the PONDR website (http://www.pondr.com). We predicted protein structure using AlphaFold2 [64].

### Protease inhibitor assay

To assess the inhibitory effects of protease inhibitor cocktail (PIC), *A. oligospora* mycelia were incubated in Potato Dextrose Broth (PDB) for 3 days. Following incubation, the mycelia were disrupted using sonication at the setting duty-cycle 50%, output control 2 (Microtip, Branson, Sonifier 250). The resulting mixture was then subjected to centrifugation at 16,000×*g* to remove the pellet. For protease inhibition, the supernatant was incubated with 1× PIC (P2714, Sigma-Aldrich) for 2 h (diluted 1:10 from the stock solution). Subsequently, 10 μL of the obtained supernatant were applied as spots onto a milk plate composed of 2% milk powder and 2% agar. The plate was incubated at 25°C for 24 h, after which imaging was performed.

To assess the potential roles of proteases in nematode colonization, *A. oligospora* culture was treated with 1× PIC for 2 h. Adult worms were then placed onto the traps and imaged using an Andor Revolution WD system with a Nikon Ti-E automatic microscope and an XON Ultra 888 EMCCD camera (Andor). Deionized water was used as the control.

## Results

### Time-course transcriptional profiling of *A. oligospora* in response to *C. elegans* reveals a highly dynamic transcriptome landscape during nematode predation

To capture the different stages of nematode predation (that is, sensing, trap formation, adhesion, penetration, and digestion [3]), we profiled the *A. oligospora* transcriptome under low nutrient conditions at various intervals following exposure to nematode prey (Fig 1A). For sequencing and all subsequent experiments, we used our newly established *A. oligospora* strain (TWF154) [8]. Principal component analysis of all transcript reads showed clustering of biological replicates for all conditions sampled (pre-exposure control, 2, 4, 10, 24, and 48 hours post-exposure (hpe) samples), certifying low variability between replicates (Fig 1B). Notably, overall transcript expression at 2 hpe and 4 hpe (earlier stages of interaction) showed tighter clustering to each other and to the pre-exposure control than to later time points (10, 24, and 48 hpe time points) (Fig 1B).

A large number of transcripts were found to be down-regulated and up-regulated in response to *C. elegans*. Specifically, using thresholds of *beta* > 1 (up-regulation) or *beta* < −1 (down-regulation), 839, 959, 1,129, 678, and 622 transcripts were up-regulated, while 1,552, 1,471, 1,760, 1,141, and 996 transcripts were down-regulated at 2 hpe, 4 hpe, 10 hpe, 24 hpe, and 48 hpe, respectively (Fig 1C and 1D). Among all the time points sampled, we observed the greatest differential expression (both up- and down-regulation) at 10 hpe, which corresponds to a period of intense trap formation and adhesion between fungal and nematode cells. Across

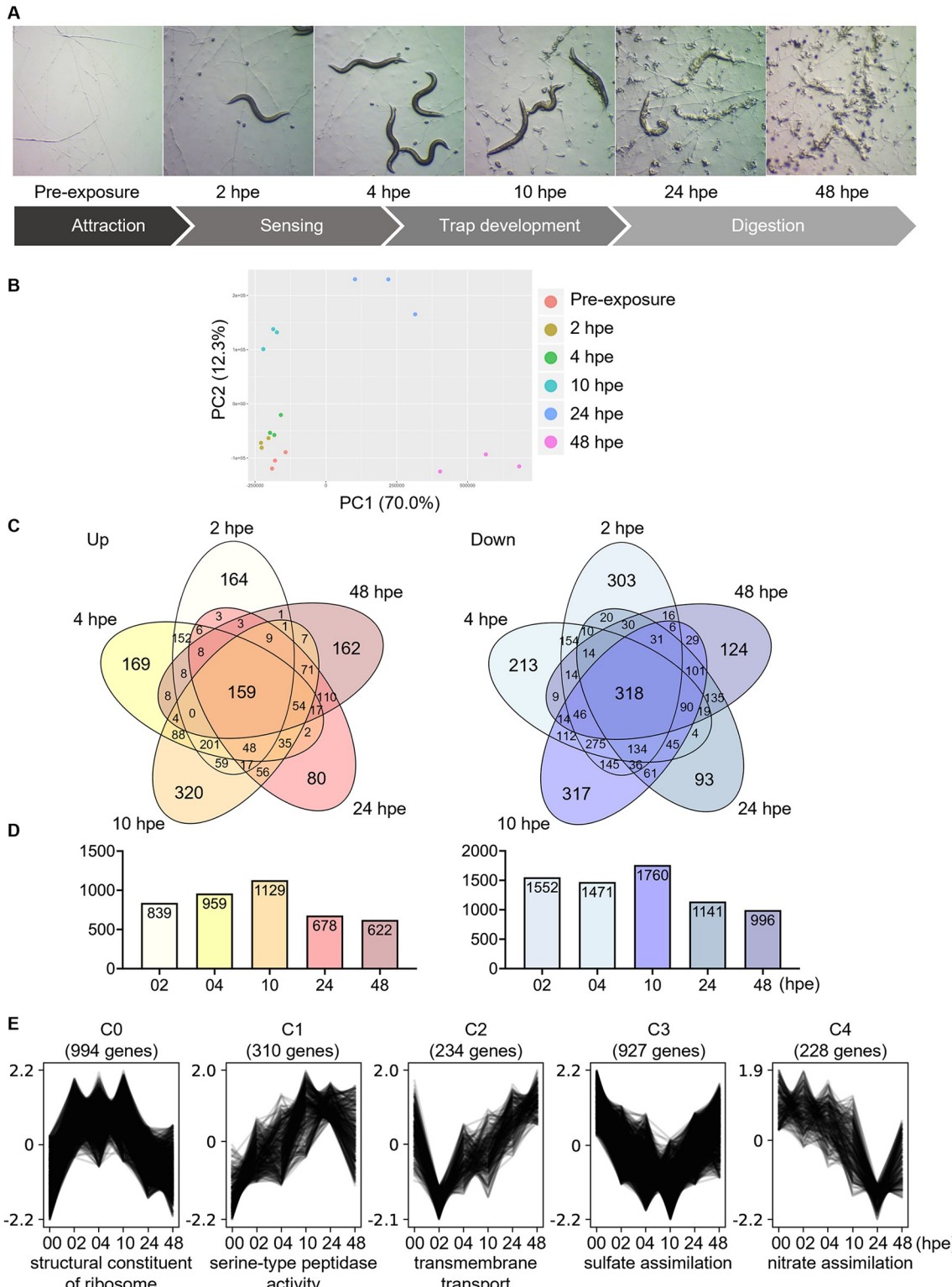

**Fig 1. *A. oligospora* responds to the presence of prey with a dynamic transcriptomic response.** (**A**) Overview of time-course RNA-seq experiment. hpe, hours post-exposure. (**B**) Principal component analysis of *A. oligospora* transcript expression upon exposure to *C. elegans* of 3 biological replicates for the indicated time points, as analyzed by RNA-seq. The data underlying this Figure can be found in S1 Data. (**C**) Venn diagrams depicting the number of *A. oligospora* transcripts that are up-regulated (left) or down-regulated (right) upon exposure to *C. elegans*. (**D**) The total number of transcripts with altered expression for each of the time points ("size of each list")

is shown. (**E**) Cluster analysis of transcript expression profiles revealed 5 common expression profiles. The 2,693 differentially expressed genes were subjected to soft clustering to identify 5 common expression profiles. The top enriched GO term for each cluster is listed, respectively. The data underlying this Figure can be found in S1 Data.

all time points, a total of 159 and 318 transcripts were up-regulated and down-regulated, respectively, and are likely to correspond to core genes required for predation (Fig 1C).

Furthermore, examination of expression levels across time for differentially expressed genes (*q*-value < 0.01 and *beta* > 1 or < −1) revealed 5 clusters of genes defined by their unique expression pattern during nematode predation (Fig 1E and S2 Table). Transcripts in the most highly occupied cluster ("cluster C0," 994 genes) showed a quick increase in expression upon exposure to *C. elegans*, sustained expression until 10 hpe, and a subsequent decrease from 24 hpe to 48 hpe (Fig 1E). This cluster was enriched in transcripts related to ribosomal biogenesis, nucleosome assembly, and translation (Fig 1E). Showing a reciprocal expression pattern to C0, cluster C3 was the second most highly occupied group, encompassing 927 transcripts. This cluster was enriched in transcripts with functions in oxidation–reduction processes that harbor metabolism, stress response, and enzymatic activity (Fig 1E). Cluster C1 transcripts (310), which showed gradual increases in expression that peaked at 10 hpe, were enriched for genes containing signal peptides. Interestingly, we also observed a strong overrepresentation of genes belonging to the DUF3129 (domain of unknown function) protein family (Fig 1E), also known as Egh16-like virulence factors [65–67]. Cluster C2 (234 transcripts) expression was significantly reduced upon contact with nematodes, followed by gradual restoration of expression to pre-exposure levels by the terminally sampled time point. This cluster primarily comprises genes involved in transmembrane transport, as well as N,N-dimethylaniline monooxygenase activity (Fig 1E). Cluster C4 transcripts (228) were characterized by a sharp decrease in expression at 24 hpe. Interestingly, C4 transcripts were found to be involved in nitrogen metabolism (specifically the conversion of nitrogen-containing compounds into usable nitrogen forms) and/or to contain domains bearing nitrate assimilation and oxidoreductase activity. Altogether, these data indicate that NTF react to the presence of nematode prey with a very rich and dynamic transcriptomic response.

## Up-regulation of ribosome biogenesis and nuclear division during trap morphogenesis

To gain further insights into trap morphogenesis, we conducted Gene Ontology (GO) enrichment analysis in *A. oligospora* samples collected between 2 and 10 hpe to *C. elegans*. Analysis revealed "structural constituent of ribosome" (GO:0003735), "rRNA processing" (GO:0006364), "nucleolus" (GO:0005730), and "nucleosome" (GO:0000786) as the top 4 most highly enriched GO terms, indicating a strong up-regulation of ribosome biogenesis and protein synthesis during the initial hours of exposure to prey (Fig 2A). These results suggest that, despite nutrient deprivation, fungal cells can ramp up ribosome biogenesis machinery in preparation for trap development upon sensing prey. Ribosome biosynthesis is known to be regulated by the TOR signaling pathway [68,69]. Thus, to examine the effect of ribosome biosynthesis on trap morphogenesis, we utilized rapamycin to inhibit TOR activity. Supplementation of rapamycin to LNM yielded a significant reduction in the number of traps induced by *C. elegans* and an increase in the *C. elegans* survival rate (Fig 2B and 2C), demonstrating that up-regulation of the ribosome biogenesis pathway is crucial for trap development. Furthermore, using a strain expressing histone H2B::mCherry to determine nuclei number in fungal cells, we observed dramatically more nuclei in trap cells than in vegetative hyphae (an average of 10 versus 2 nuclei) (Fig 2E). Consistent with this observation, the GO terms

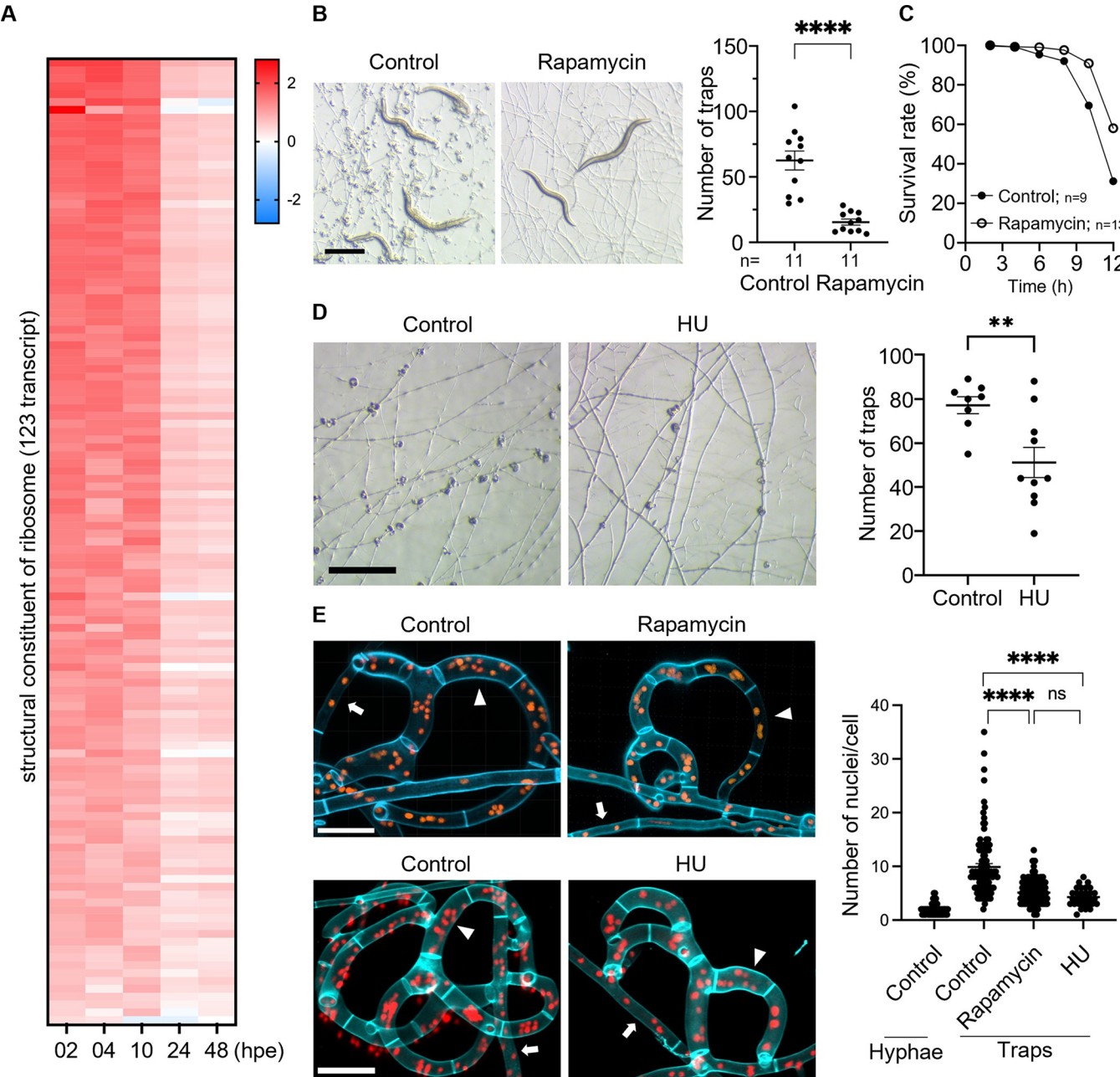

**Fig 2. Ribosome biosynthesis is up-regulated in early time point.** (**A**) Expression profile of structural constituents of ribosome upon exposure to *C. elegans*. The data underlying this Figure can be found in S1 Data. (**B**) Trap induction and quantification of WT *A. oligospora* grown on LNM and LNM with 4 μg/ml rapamycin (scale bar, 500 μm). The data underlying this Figure can be found in S1 Data. (**C**) Survival rate of *C. elegans* on WT *A. oligospora* grown on LNM and LNM with 4 μg/ml rapamycin. The data underlying this Figure can be found in S1 Data. (**D**) Trap induction and quantification of WT *A. oligospora* on LNM and LNM with 200 mM HU (scale bar, 500 μm). The data underlying this Figure can be found in S1 Data. (**E**) Close-up images and quantification of the nuclei of the H2B::mCherry strain in the traps of *A. oligospora* grown on LNM and LNM with 4 μg/ml rapamycin or 200 mM HU (scale bar, 20 μm). The arrowhead indicates the trap cell; the arrow indicates vegetative hyphae. The data underlying this Figure can be found in S1 Data. hpe, hours post-exposure; HU, hydroxyurea; LNM, low-nutrient medium; WT, wild-type.

"nucleolus" (GO:0005730) and "nucleosome" (GO:0000786) are highly enriched in 2 to 10 hpe differentially expressed genes (S1 Fig). In the presence of rapamycin, nuclei number per trap cell was reduced to approximately 5 nuclei (Fig 2E).

Given the nearly 5-fold increase in DNA content in trap cells over vegetative hyphae, we next asked whether inhibition of DNA replication might affect trap morphogenesis. We applied the known DNA synthesis inhibitor HU [70,71] to fungal cells induced with nematodes and found that HU treatment moderately inhibited trap development and reduced the number of nuclei to approximately 4 nuclei per trap cell (Fig 2D and 2E). Taken together, these findings suggest that, under nutrient deprivation, *A. oligospora* up-regulates ribosome biogenesis, translation, and nuclear division to prepare for trap development in response to nematode signals.

## *A. oligospora* exhibits strong reliance on secretion-related genes during interactions with *C. elegans*

Strikingly, in our functional analysis, we observed a strong enrichment of differentially expressed genes with functions related to protein secretion. Indeed, applying a stringent threshold of $beta > 2$ nevertheless resulted in highly significant ($q$-value = $1.7 \times 10^{-101}$) enrichment of the signal peptide category "SignalP-noTM" at 10 hpe (S3 Table). This suggests that the *A. oligospora* response to *C. elegans* involves vigorous induction of protein secretion.

To better understand the role of protein secretion in the fungal prey response, we predicted the *A. oligospora* secretome using a combination of outputs from SignalP [43], Phobius [44], TMHMM [45], and WolfPSort [46]. A total of 1,084 peptides were predicted to be secreted in *A. oligospora*, representing 8.25% of the transcriptome. From this list, 549 transcripts (51% of 1,084) showed altered expression in our RNA-seq dataset ($q$-value < 0.01; $beta > 1$ or $< -1$) (Fig 3A). Expression pattern and functional enrichment analysis of the 549 transcripts revealed that a large proportion were up-regulated at 4 to 10 hpe (Fig 3B) and displayed an overrepresentation for a few select protein domains (S4 Table). Within the predicted secretome, the "DUF3129," "Glycoside hydrolase superfamily," "Peptidase S8/S53," and "Carbohydrate-binding WSC" protein domains were highly enriched (S4 Table).

Work in pathogenic fungi has demonstrated an important role for effector proteins in infection [72–74]. In this context, we therefore reasoned that effector proteins might play a role in nematode predation by manipulating nematode cellular processes or influencing behavior during trapping. To investigate this hypothesis, we used the predicted secretome to identify putative effectors in *A. oligospora*, resulting in a list of 417 predicted transcripts encoding effectors (representing 38.5% of the secretome) (Table 2). Two hundred and nine predicted effectors showed differential expression in the presence of *C. elegans* ($q$-value < 0.01; $beta > 1$ or $< -1$; Fig 3C), with the majority of the predicted effectors induced at 4 or 10 hpe (Fig 3D). While 259 predicted effectors lacked Pfam domains, functional enrichment analysis of the differentially expressed predicted effectors encoding known domains again revealed the prevalence of the "DUF3129" domain, as well as "peptidase M43," "cysteine-rich secretory protein-related," and "ribonuclease T2-like" domains.

To further investigate the role of secretion in NTF, we investigated how disruption of protein secretion might affect *A. oligospora* predation by leveraging a genetic mutant deficient in the highly conserved syntaxin gene EYR41_000055. Syntaxin belongs to the t-SNARE family, is known to regulate the fusion of secretory vesicles with the plasma membrane during the late stages of the secretory pathway, and has been shown to enhance protein secretion when overexpressed in the budding yeast *Saccharomyces cerevisiae* [75,76]. We generated a targeted gene deletion mutant in the plasma membrane t-SNARE protein *sso2* in the *ku70* background, which facilitates homologous recombination in *A. oligospora*. While the *sso2* mutant exhibited no defects in trap development (Fig 3E and 3F), we found that *sso2* mutant traps exhibited reduced functionality compared to wild-type traps. Specifically, 70% of nematodes were able

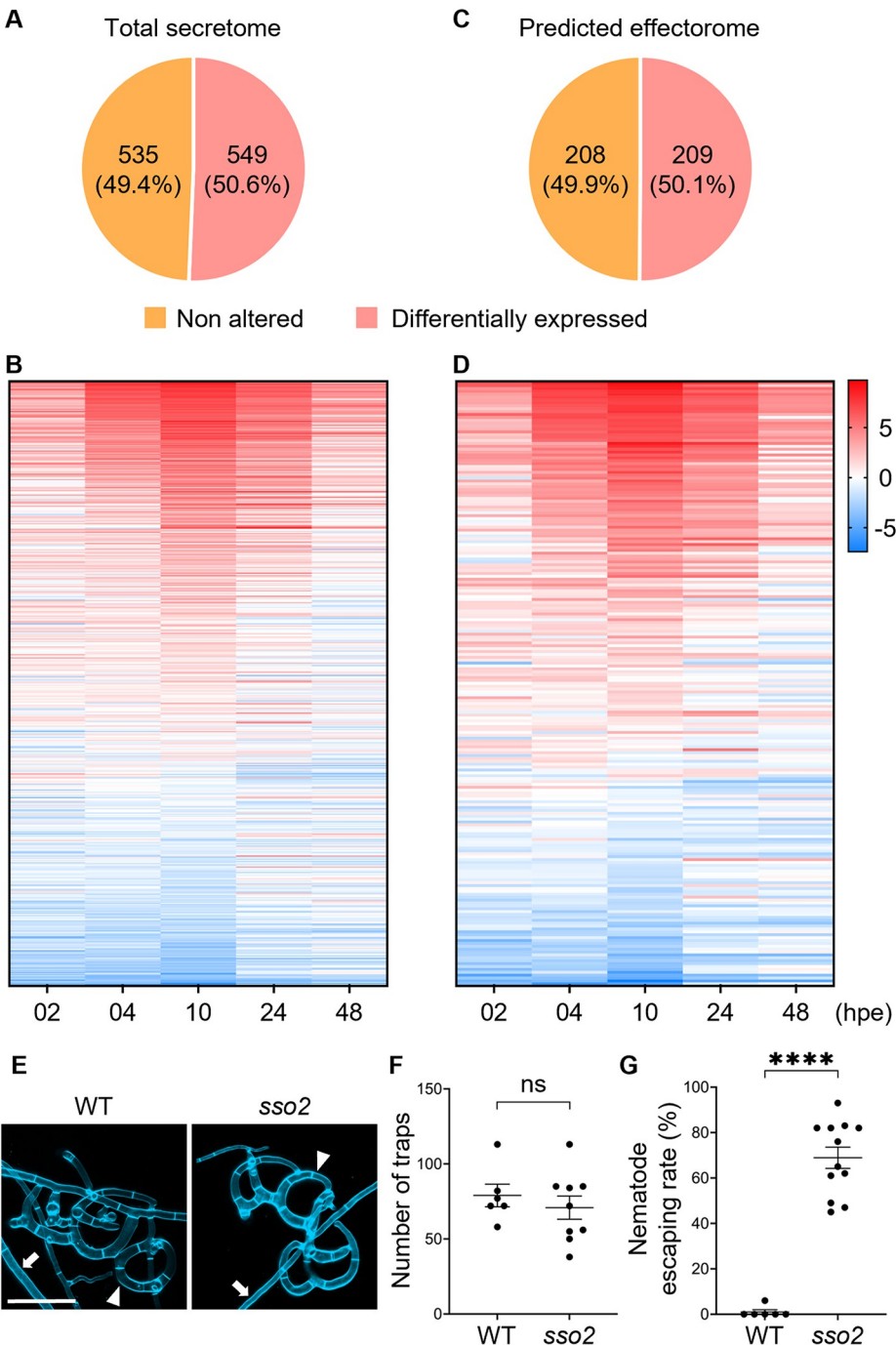

**Fig 3. *A. oligospora*'s response to *C. elegans* is dependent on protein secretion.** (**A-D**) Percentage and expression pattern analysis of the predicted secretome (**A**, **B**) and effectorome (**C**, **D**) that showed differential expression upon exposure to *C. elegans*. The data underlying this Figure can be found in S1 Data. (**E**) High-resolution image of traps stained by SR2200. Images were taken 24 h after induction (scale bar, 50 μm). The arrowhead indicates the trap cell; the arrow indicates vegetative hyphae. (**F**, **G**) Quantification of trap numbers (**F**) induced by exposure to 30 *C. elegans* nematodes and nematode escaping rate. (**G**) Among trapping of 50 *C. elegans* (mean ± SEM). The data underlying this Figure can be found in S1 Data.

**Table 2. Number of secretome and effectorome in *A. oligospora*.**

|  | Number of transcripts | % |
|---|---|---|
| Total proteins in secretome | 1,084 |  |
| Predicted effectors | 417 | 38.5 |
| Predicted cytoplasmic effectors | 114 | 10.5 |
| Predicted apoplastic effectors | 303 | 28.0 |
| Non-effectors | 667 | 61.5 |

to escape *sso2* traps, in contrast to the nearly 0% escape rate from wild-type *A. oligospora* traps (Fig 3G and S1 and S2 Movies). These data demonstrate that protein transport and secretion, likely of nematode-adhesive molecules, is important for trap function.

## Members of the DUF3129 gene family are expanded in the genome of nematode-trapping fungi

Having identified significant enrichment of the DUF3129 domain in our predicted secretome and effectorome datasets, we further probed the role of DUF3129 domain-containing genes in *A. oligospora*. A specific examination of this family in our RNA-sequencing demonstrated that several members of the family were prominently induced in *A. oligospora* when exposed to nematodes (Fig 4A). Notably, among the 27 DUF3129 transcripts, 15 exhibited peak induction at 10 hpe to *C. elegans* (Fig 4A), suggesting a potential role of DUF3129 proteins in trap development. *A. oligospora* genome encodes 27 DUF3129 domain-containing genes, and except for one, all of these DUF3129 domain genes are predicted to contain a signal peptide by SignalP, indicating that this domain likely defines an expanded family of secreted proteins. Furthermore, approximately 40% (11 genes) of the family members were identified as putative effectors in our analysis. We further examined the distribution of the DUF3129 genes in the *A. oligospora* genome and found them scattered across 8 chromosomes without apparent genomic clustering (S2 Fig).

We next looked for DUF3129 genes distribution within the fungal kingdom by scanning 2,504 fungal genomes comprised in the mycocosm fungal database [51] and 8 genomes of NTF (S5 Table). Interestingly, the majority of species (approximately 99%) contain fewer than 5 DUF3129 genes in their genomes, whereas this number dramatically increased to more than 20 in several species of NTF (Fig 4B). In order to verify and expand these initial findings, we performed orthogroup analysis of the DUF3129 genes in 19 selected high-quality genomes of diverse fungi (S5 Table). A total of 108 DUF3129 genes were distributed across 12 of these genomes. Of these DUF3129 genes, 86 were grouped into 23 orthogroups (Fig 4C). As shown in Fig 4C, the majority of the *A. oligospora* DUF3129 genes were specific to the *Arthrobotrys* genus. Additionally, our analysis revealed that the family of genes containing DUF3129 has expanded specifically within the NTF. This association implies that their expansion, followed by sequence divergence, likely contributed to the unique biology of the NTF species (S3 Fig). Our investigation into gain/loss patterns indicated that most of the duplication events involving DUF3129 genes occurred within Ascomycete NTF clades. Notably, even within the *A. oligospora* branch, we detected an additional 15 DUF3129 gene duplication events. This demonstrates that gene expansion has even occurred after speciation from *A. vermicola*. On the other hand, only a few duplication events were observed in non-NTF. The number of DUF3129 genes within plants, insects, humans, and other nematode pathogens remained limited, ranging from 0 to 6. These findings further emphasize an evolutionary expansion of DUF3129 genes, that is potentially linked to the predatory lifestyle of NTF.

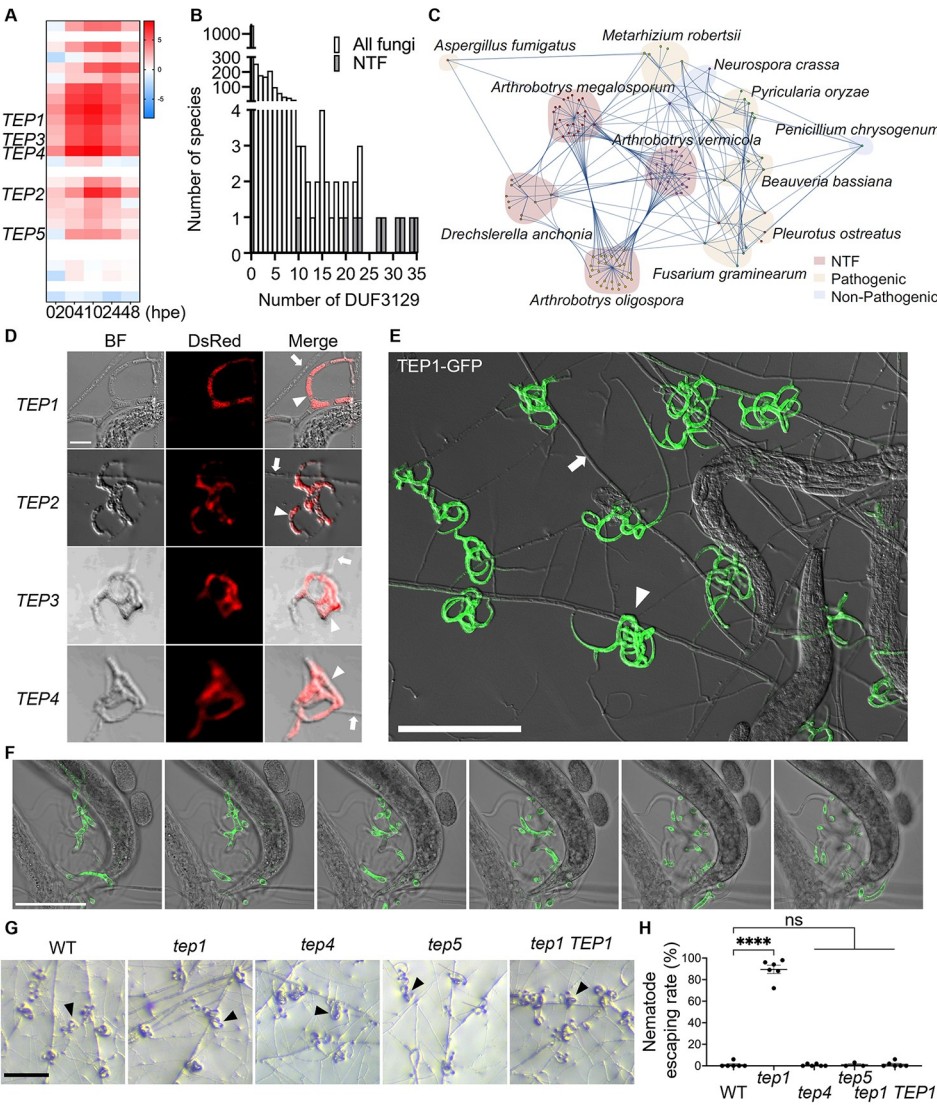

**Fig 4. The TEP family is an important component of the *A. oligospora* response to *C. elegans*.** (**A**) Expression profiles of TEP transcripts upon exposure to *C. elegans*. The data underlying this Figure can be found in S1 Data. (**B**) Histogram depicting the number of genes containing the DUF3129 domain encoded in the 2,512 fungal genomes (S5 Table). The data underlying this Figure can be found in S1 Data. (**C**) Orthogroup analysis cloud map. Each dot represents a DUF3129 protein from the indicated species, with lines connecting proteins belonging to the same orthogroup. The data underlying this Figure can be found in S1 Data. (**D**) High-resolution confocal imaging of 4 highly expressed *TEP* transcripts in WT traps. The WT strain was induced by nematodes for 24 h. Hairpins were conjugated with Alexa Fluor 647, 594, 546, and 514 (red) (scale bar, 20 μm). The arrowhead indicates the trap cell; the arrow indicates vegetative hyphae. (**E**) An image of trap induction of the TEP1-GFP strain (scale bar, 200 μm). The arrowhead indicates the trap cell; the arrow indicates vegetative hyphae. (**F**) Confocal images of the localization of TEP1-GFP (scale bar, 100 μm). (**G**) Images of traps of the WT, *tep1*, *tep4*, *tep5*, and *tep1 TEP1* following a 6-h exposure to 30 *C. elegans* nematodes; the images were captured 24 h later (scale bar, 200 μm). The arrowhead indicates the trap cell. (**H**) Quantification of the nematode escaping rate of 50 *C. elegans* (mean ± SEM). The data underlying this Figure can be found in S1 Data. BF, bright field; hpe, hours post-exposure; NTF, nematode-trapping fungi; TEP, Trap Enriched Protein; WT, wild-type.

To further investigate the potential roles of the DUF3129 domain-containing proteins, we first determined their expression pattern by *in situ* hybridization chain reaction (HCR). Probes for the 4 most highly expressed DUF3129 family genes, EYR41_003105, EYR41_001312,

EYR41_005998, and EYR41_0011738, were designed and hybridized to *A. oligospora* culture at 10 hpe to *C. elegans*, where they showed high enrichment in trap cells (Fig 4D). We then investigated the expression and localization of the most highly expressed member, EYR41_003105, through fusion to GFP. EYR41_003105 was prominently enriched in traps (Fig 4E) and localized on the surface of the trap cell (Fig 4F), while showing weak or no signal in regular hyphae (Fig 4E). Moreover, live imaging revealed the flow of EYR41_003105 from hyphae into traps (S3 Movie). These results provide strong evidence that DUF3129 family genes accumulate in traps at both the transcriptional and translational level. Therefore, we named this class of highly expressed DUF3129 members Trap Enriched Protein (TEP) genes.

Having established the expression and localization of TEP genes to traps, we next examined their involvement in trap development and function. Targeted gene deletion mutants of EYR41_003105 (*tep1)*, EYR41_0011738 (*tep4)*, and EYR41_009694 (*tep5)* were generated and assessed for their ability to develop traps in response to *C. elegans*. We found that all mutants developed comparable number of traps to wild type in response to *C. elegans*, suggesting that these TEP genes do not play a role in nematode-sensing and trap morphogenesis (Figs 4G and S4). Instead, given TEP protein localization to the surface of trap cells, we hypothesized that TEPs may be critical for the function of the traps themselves. Adding *C. elegans* onto the traps of wild-type *A. oligospora* leads to their immediate capture. However, in the *tep1* mutant, only 10% of nematodes were captured after 10 min of exposure to the traps, with trap efficacy restored upon reconstitution of TEP1 (Fig 4H and S1 and S4 Movies), indicating a potential role for TEP1 in trap adhesion needed to catch *C. elegans*.

Despite the fact that all 27 *A. oligospora* DUF3129 proteins share a conserved domain in the central portion of their sequence, a sequence alignment of the 4 most strongly expressed DUF3129 genes demonstrates high sequence diversity among these proteins (S5 Fig).

Structural predictions of TEP protein sequences indicated that the C-terminus region of the TEPs exhibit a high degree of disordered (S6 Fig). Because IDRs of proteins can drive liquid–liquid phase separation (LLPS), or the compartmentalization of proteins into "membraneless organelles" [77], it is possible that TEPs form liquid condensates in traps (S6 Fig). Aggregation of TEPs and other proteins might be required for trap adhesion. Taken together, these data support the idea that the DUF3129 family may have expanded in NTF and contribute to the function of traps.

## Late time point induced protease and its potential role in digesting nematodes

At mid to late time points, we observed hyphal growth inside nematodes following penetration of the cuticle, corresponding to digestion. Hypothesizing proteases may be important for nematode digestion, we searched the transcriptome for proteases predicted by the MEROPs peptidase database. We identified a total of 323 protease-encoding genes; of these, 27, 56, 87, 123, and 19 genes were characterized as encoding aspartic, cysteine, metallo, serine, and threonine peptidases, respectively. Notably, genes encoding each class of protease (15 aspartic, 14 cysteine, 39 metallo, 62 serine, and 3 threonine) were differentially expressed in response to nematodes (Fig 5A). Consistent with a role in digestion, many of the protease genes were highly expressed at late time points when nematodes have already been captured (Fig 5B).

Given the large number of predicted proteases and potential for redundancy in digestive function, we reasoned that mutation of a single protease might not be sufficient to see obvious changes in digestion of nematodes. Therefore, to investigate the potential involvement of proteases in hyphal colonization and prey digestion, we utilized a protease inhibitor cocktail (PIC) to inhibit overall protease activity after trap morphogenesis and measured hyphal growth

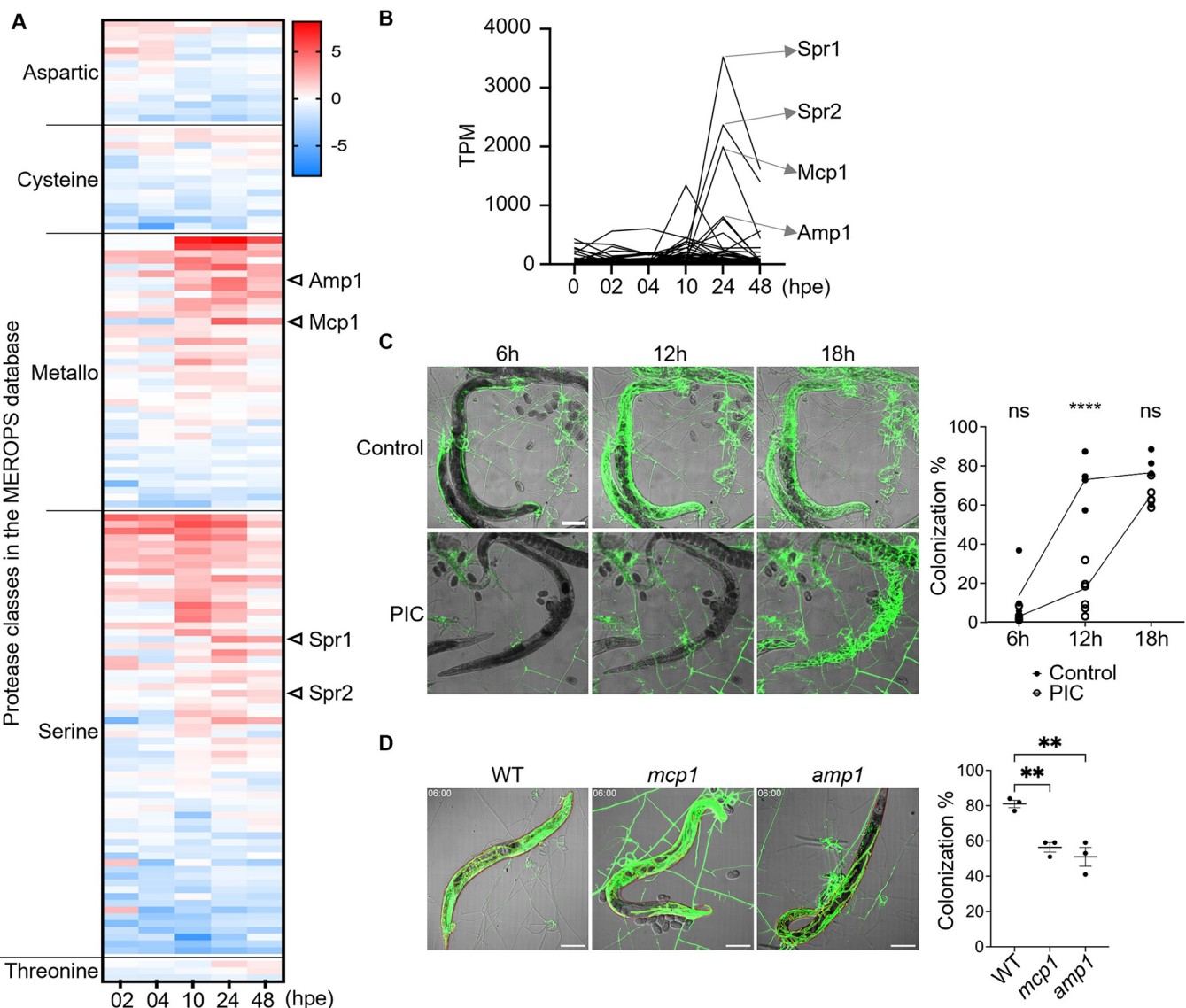

**Fig 5. Late time point induced protease and its potential role in digesting nematodes.** (**A**) Expression profiles of differentially expressed protease transcripts upon exposure to *C. elegans*. The data underlying this Figure can be found in S1 Data. (**B**) TPM of the differentially expressed proteases. Arrows indicate the 4 highly up-regulated proteases. The data underlying this Figure can be found in S1 Data. (**C**) Image and quantification of hyphal colonization after penetration for 6–18 h and treatment with 1× PIC (scale bar, 100 μm). The data underlying this Figure can be found in S1 Data. (**D**) Image and quantification of hyphal colonization in *C. elegans* after being captured by traps for 6 h in WT, *mcp1*, and *amp1* (scale bar, 100 μm). The data underlying this Figure can be found in S1 Data. hpe, hours post-exposure; PIC, protease inhibitor cocktail; TPM, Transcript Per Million; WT, wild type.

inside nematode bodies. Visualization and quantification of hyphal growth using cytosolic GFP expressed in the NTF revealed delayed hyphal colonization with PIC treatment (Fig 5C), suggesting an important role of proteases in both hyphal growth and prey digestion. Additionally, as a control, we conducted a milk plate assay, assessing the proteolytic activity of *A. oligospora* with or without PIC treatment and demonstrated that inhibitor-treated samples showed a smaller clear zone, but some residual enzymatic activity persisted despite the treatment (S7 Fig).

We next investigated individual proteases in the fungus. From our RNA-seq dataset, we identified 2 metallopeptidase genes that were highly differentially expressed at 24 hpe (Fig 5A and 5B). To investigate the involvement of metallopeptidases in nematode digestion, we

generated targeted gene deletions of *mep1* (MEtalloPeptidase 1) and *amp1* (AMinoPeptidase 1) in the *ku70* background. Both mutants were able to generate traps in response to nematodes, but as visualized by cytosolic GFP displayed minor defects in hyphal colonization (Fig 5D). Together, our findings suggest that proteases, specifically metallopeptidases, play a role in facilitating invasive hyphal growth and colonization of nematodes.

## Discussion

The molecular basis underlying the relationship between NTF and nematodes remains largely uncharacterized. Here, we used the *A. oligospora*–*C. elegans* model and a set of RNA-seq time course experiments to uncover genes and pathways that play roles at various stages of the predator–prey interaction.

At early time points, we observed an up-regulation of a large proportion of ribosome biosynthesis genes and an increase in nuclei number in trap cells (Fig 2A and 2E). Functional analysis further confirmed that ribosome biosynthesis and DNA replication play a role in trap development (Fig 2B–2D). Half of the identified secretome of *A. oligospora* showed altered expression upon exposure to *C. elegans* (Fig 3). Thus, *A. oligospora* activates a strong secretory response when exposed to prey under starvation conditions. Altogether, these data suggest that such increases in DNA content and translational capacity are likely necessary to support increased protein secretion. In fruit fly, salivary gland cells, which have high secretion demands, also exhibit similar features of up-regulation of ribosomal protein genes and nuclear division [78,79]. Therefore, enhanced translation capacity and larger cell size may be conserved features of cells needing to enhance secretion capacity.

Given the large size of the *A. oligospora* secretome and effectorome and the differential expression of these factors in response to *C. elegans* (Fig 3), we hypothesize that effector proteins play a crucial role in the predation process. Analogous work in pathogenic fungi, which show that effector proteins can manipulate the host's immune system and facilitate fungal infection [72–74], support this hypothesis. For example, genomic sequencing in the corn smut fungus *Ustilago maydis* identified 298 genes encoding secreted proteins with unknown function, with many of these genes clustered in the *U. maydis* genome. Subsequent deletion of these genomic clusters led to compromised fungal proliferation in the host [80], suggesting that these genes correspond to effector proteins required for *U. maydis* pathogenicity. Similarly, transcriptional profiling of the rice blast fungus, *Magnaporthe oryzae*, during its entire plant-associated development, identified 32 MEP effector genes required for rice infection [74]. Specifically, MEPs were found to target the cytoplasm of rice cells via the biotrophic interfacial complex and employ an unconventional secretory pathway. In the wheat pathogen *Zymoseptoria tritici*, the effector protein Zt6 is responsible for wheat cell death and defense against other microbial competitors [81]. Together, these studies highlight the importance of fungal effector proteins in diverse aspects of fungal pathogenicity, including proliferation, tumor formation, and niche protection from other microbes. In this study, we identified 417 predicted effectors in *A. oligospora*, with over half of them showing differential expression in response to *C. elegans*. Furthermore, more than 70% of the predicted effectors were also predicted to be apoplastic effectors (Table 2), suggesting their potential role against antifungal proteins [73]. Given that *C. elegans* produces antimicrobial peptides (AMPs) in response to fungal infection [82], the *A. oligospora* apoplastic effectors we identified here may be important for combating nematode-secreted AMPs and ensuring successful predation. Nevertheless, further characterization of these effectors and their role in pathogenesis is needed and presents a fascinating avenue for further investigation.

While searching for transcripts with strong induction upon exposure to *C. elegans*, we found a particularly strong enrichment of members of the fungal-specific DUF3129 family. Interestingly, we found that this gene family has undergone remarkable expansion and diversification in *A. oligospora*, with 27 predicted members. We hypothesize that the evolution and expansion of the DUF3129 family in specific fungal lineages, resulting in dramatic size differences between family members, may be attributed to pressures to adapt to a wide diversity of nematode species, as has previously been suggested for virulent microsporidia [83]. Furthermore, our data suggest that DUF3129 genes, specifically TEP genes, are important for trap adhesion and predation (Fig 4). Work in fungal pathogen–host systems have also implicated DUF3129 genes in fungal pathogen virulence. In the rice blast fungus *M. grisea*, 2 DUF3129 genes were shown to be expressed in appressoria and involved in host penetration and lesion development [84]. In *Metarhizium acridum*, 1 DUF3129 gene was shown to be involved in insect penetration [85]. In *M. robertsii*, DUF3129 proteins were localized to cellular lipid droplets, and null mutants of DUF3129 genes showed impaired virulence in topical infections of insects but not in injections of insects, again suggesting a role in penetrating the host cuticle [86]. Additionally, work in several fungal pathogens suggests that DUF3129 proteins act downstream of the mitogen-activated protein kinase (MAPK) family and its putative substrate, Ste12 [86]. Following phosphorylation by the MAPK gene Pmk1, the transcription factor Ste12 is hypothesized to drive the expression of DUF3129 genes, such as the Gas1-like genes in *M. acridum* [87]. In our previous work, we have demonstrated that the conserved pheromone MAPK signaling pathway and the downstream transcription factor Ste12 are required for nematode sensing and trap development in *A. oligospora* [18]. Furthermore, RNA sequencing analysis in a *ste12* mutant identifies the TEP family genes as part of the Ste12 regulon (S8 Fig). Together, these data suggest that regulation of DUF3129 effector genes by the MAPK pathway and Ste12 may be a conserved feature of fungal pathogens.

While this and previous studies suggest that DUF3129 genes are important for fungal life strategies, whether predation or parasitism, the mechanism of how TEP genes mediate trap adhesion remains to be investigated. However, other studies lead us to propose 2 possible hypotheses for the biochemical function of TEPs in trap cells. First, TEPs may act as the adhesins that directly adhere to the nematode cuticle. In the wood decay fungus *Phanerochaete chrysosporium*, the DUF3129 protein OSIP1 is critical for extracellular matrix (ECM) formation. Intriguingly, the heterologously expressed protein has the ability to self-assemble into fibers and form a gel that shields fungal cells from toxins and antifungal drugs [88]. Similarly, we have predicted IDRs in TEP genes (S6 Fig). Therefore, similar to *P. chrysosporium* OSIP1, TEP genes may self-assemble to form an ECM that makes traps stickier and better at trapping nematodes. Alternatively, TEPs may exert enzymatic activity on the fungal cell wall directly to convert vegetative hyphae into adhesive traps. Recently, a comprehensive protein sequence analysis across both Eukaryotes and Bacteria suggested that the DUF3129 gene family may be distantly related to lytic polysaccharide monooxygenases (LPMOs) [89], which catalyze the cleavage, oxidation, and activation of cell wall components such as cellulose, xylan, and chitin [89]. Therefore, we might envision that TEP genes play a direct enzymatic role in modifying the fungal cell wall to confer adhesion. Further characterization of the molecular function of TEPs, including investigating whether they function directly as adhesins or are involved in the degradation of cell wall components, will provide insights into the mechanism of trap adhesion between fungal predators and their nematode prey.

NTF employ various mechanisms to capture and digest nematodes, including the production of specialized enzymes such as proteases and peptidases. Studies have suggested that serine proteases and metalloproteases are involved in immobilizing nematodes post-capture [90]. One of the extracellular serine proteases, PII (X94121/AOL_s00076g4), is capable of degrading

the nematode cuticle [91]. Despite being inhibited by various serine protease inhibitors [92], the specific role of PII in the digestion phase remains unclear. Our results here showed that the inhibition of protease activity resulted in delayed invasive hyphae penetration and impaired nematode colonization. Furthermore, we observed that mutants in 2 highly expressed metallo-peptidases showed a minor defect in the colonization of nematode bodies after penetration (Fig 5). Together, these finding suggest that protease genes, including metallopeptidases, are involved in the nematode digestion process and merit further characterization. Lastly, while our current findings provide valuable insights into the fungus' response to starvation and predation, the role of autophagy in nutrient scavenging remains an intriguing avenue for future research. We did not observe an obvious regulatory pattern of the autophagy-related genes in our study, but previous work has suggested that certain autophagy-related genes may regulate trap formation (S9 Fig) [93–96].

In summary, our time-course transcriptional profiling study provides insight into the complex mechanisms underlying the interaction between *A. oligospora* and *C. elegans*. Namely, we have identified that upon nematode exposure, *A. oligospora* first up-regulates ribosome biosynthesis and DNA replication. Subsequently, mobilization of large portions of the total secretome and effectorome assists trap development and function, with TEP genes especially critical for nematode adhesion. Lastly, after penetration into the host, *A. oligospora* relies on the action of peptidases to digest nematodes. This work lays a solid foundation for future investigation of the molecular mechanisms underlying predator–prey interactions between NTF and nematodes and allows parallel comparison with other pathogenic fungi and their interactions with plant or animal hosts.

## Supporting information

**S1 Data. This spreadsheet contains the data presented in the figures.**
(XLSX)

**S1 Table. Primers used in this study.**
(XLSX)

**S2 Table. Functional enrichment analysis of the different clusters of genes.**
(XLSX)

**S3 Table. Functional enrichment analysis of the up- and down-regulated transcripts.**
(XLSX)

**S4 Table. Functional enrichment analyses of the components of the secretome that showed differential expression upon exposure to *C. elegans*.**
(XLSX)

**S5 Table. Orthology analysis of DUF3129 family.**
(XLSX)

**S1 Fig. Functional enrichment analysis of up- and down-regulated transcripts.** Gene Ontology categories for each time point with a *p*-value < 0.05. The data underlying this Figure can be found in S1 Data.
(TIF)

**S2 Fig. Genomic location of the DUF3129 containing genes of *A. oligospora* TWF154.**
(TIF)

**S3 Fig. Orthogroup analysis of gene gain and loss of DUF3129 gene family.** The numbers indicate the count of DUF3129 genes. The data underlying this Figure can be found in S1

Data.
(TIF)

**S4 Fig. Quantification of the number of traps of the WT, *tep1*, *tep4*, *tep5*, and *tep1 TEP1* induced by the presence of 30 *C. elegans* (mean ± SEM).** The data underlying this Figure can be found in S1 Data.
(TIF)

**S5 Fig. Amino acid alignment of the top 4 expressed TEP proteins of *A. oligospora*.**
(TIF)

**S6 Fig. Sequence analyses of the top 4 TEP proteins.** (**A**, **B**) Gene model (**A**) and protein structure prediction (**B**) of the top 4 expressed TEP proteins of *A. oligospora*. (**C**) The confocal image of close-up look of trap induction of the TEP1-GFP strain (scale bar, 50 μm).
(TIF)

**S7 Fig. Clear zone analysis on milk plate demonstrating the impact of PIC treatment.** A comparison between samples subjected to PIC treatment (left) and samples without PIC treatment (right), illustrating the presence of clear zones.
(TIF)

**S8 Fig. TPM of the *TEP1*, *TEP2*, *TEP3*, and *TEP4* in WT and *ste12* with or without induction by *C. elegans* (mean ± SEM).** The data underlying this Figure can be found in S1 Data.
(TIF)

**S9 Fig. Expression pattern of autophagy-related genes in response to *C. elegans*.** The data underlying this Figure can be found in S1 Data.
(TIFF)

**S1 Movie. WT *A. oligospora* capturing nematode.**
(MP4)

**S2 Movie. *sso2 A. oligospora* capturing nematode.**
(MP4)

**S3 Movie. Live imaging of *A. oligospora* strain expression TEP1-GFP.**
(AVI)

**S4 Movie. *tep1 A. oligospora* capturing nematode.**
(MP4)

## Acknowledgments

The authors would like to thank the IMB Genomics Core and Imaging Core for technical assistance for conducting Illumina sequencing and imaging, and Ling-Mei Hsu and A-Mei Yang for their technical assistance. The plasmid containing GFP was kindly provided by Dr. Reinhard Fischer. We also thank supports from the EMBO YIP and GIN programs.

## Author Contributions

**Conceptualization:** Hung-Che Lin, Yen-Ping Hsueh.

**Data curation:** Hung-Che Lin, Guillermo Vidal-Diez de Ulzurrun, Yen-Ping Hsueh.

**Formal analysis:** Hung-Che Lin, Guillermo Vidal-Diez de Ulzurrun.

**Funding acquisition:** Erich M. Schwarz, Yen-Ping Hsueh.

**Investigation:** Hung-Che Lin, Guillermo Vidal-Diez de Ulzurrun, Sheng-An Chen, Ching-Ting Yang, Tomoyo Iizuka, Tsung-Yu Huang, Chih-Yen Kuo, A. Pedro Gonçalves, Siou-Ying Lin, Yu-Chu Chang, Yen-Ping Hsueh.

**Methodology:** Hung-Che Lin, Guillermo Vidal-Diez de Ulzurrun, Sheng-An Chen, Ching-Ting Yang, Tomoyo Iizuka, Tsung-Yu Huang, Chih-Yen Kuo, A. Pedro Gonçalves, Siou-Ying Lin, Yu-Chu Chang, Jason E. Stajich, Erich M. Schwarz.

**Project administration:** Yen-Ping Hsueh.

**Resources:** Erich M. Schwarz, Yen-Ping Hsueh.

**Software:** Guillermo Vidal-Diez de Ulzurrun, Tomoyo Iizuka, Jason E. Stajich, Erich M. Schwarz.

**Supervision:** Yen-Ping Hsueh.

**Writing – original draft:** Hung-Che Lin, Rebecca J. Tay, A. Pedro Gonçalves.

**Writing – review & editing:** Guillermo Vidal-Diez de Ulzurrun, Rebecca J. Tay, Erich M. Schwarz, Yen-Ping Hsueh.

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
