## [Editor Report · Decision Letter 0]

7 Jun 2023

Dear Dr Hsueh, 

Thank you for submitting your manuscript entitled "Dynamic transcriptional profiling in a nematode-trapping fungus reveals key processes required for various stages of fungal carnivory" for consideration as a Research Article by PLOS Biology.

Your manuscript has now been evaluated by the PLOS Biology editorial staff, as well as by an academic editor with relevant expertise, and I'm writing to let you know that we would like to send your submission out for external peer review.

Once your full submission is complete, your paper will undergo a series of checks in preparation for peer review. After your manuscript has passed the checks it will be sent out for review. To provide the metadata for your submission, please Login to Editorial Manager (https://www.editorialmanager.com/pbiology) within two working days, i.e. by Jun 09 2023 11:59PM.

Kind regards,

Roli Roberts

Roland Roberts, PhD

Senior Editor

PLOS Biology

rroberts@plos.org

---

## [Decision Letter · Decision Letter 1]

20 Jul 2023

Dear Dr Hsueh,

Thank you for your patience while your manuscript "Dynamic transcriptional profiling in a nematode-trapping fungus reveals key processes required for various stages of fungal carnivory" was peer-reviewed at PLOS Biology. It has now been evaluated by the PLOS Biology editors, an Academic Editor with relevant expertise, and by three independent reviewers. 

You’ll see that reviewer #1 is broadly positive, but spends much of the review questioning the section that describes the evolutionary analysis of the DUF3129 family (Fig 4ABC, essentially); this s/he says is too poorly methodologically described to be able to judge its robustness, and s/he gives you clear instructions as to how to rectify this. S/he also wants clearer definitions of “secretome” and “effectome” (with the implication that these might harbour problems…). Reviewer #2 is very positive, but wonders if the limited GO annotation might skew results, and whether you could check for unanticipated effects of the inhibitors. Reviewer #3 is also very positive, but has a number of requests; these are mostly modest, but some will entail some minor experiments or analyses: quantify stages of trap development, look for autophagy-related gene involvement, check the activity of the inhibitors (as for rev #2), check for genomic clustering of DUF3129 genes, add some controls…

In light of the reviews, which you will find at the end of this email, we would like to invite you to revise the work to thoroughly address the reviewers' reports.

Given the extent of revision needed, we cannot make a decision about publication until we have seen the revised manuscript and your response to the reviewers' comments. Your revised manuscript is likely to be sent for further evaluation by all or a subset of the reviewers.

**IMPORTANT - SUBMITTING YOUR REVISION**

*Re-submission Checklist*

*Published Peer Review*

*PLOS Data Policy*

*Blot and Gel Data Policy*

Sincerely,

Roli Roberts

Roland Roberts, PhD

Senior Editor

PLOS Biology

rroberts@plos.org

REVIEWERS' COMMENTS:

Reviewer #1:

Evolution of diverse lifestyles is an exciting question. In the Ascomycetes, the nematode trapping fungi exhibit the unique ability to switch their morphology and physiology in order to trap and digest nematode prey. The authors in this manuscript used time course transcriptomics to identify gene sets that are differentially expressed at different stages of the trapping. They identified at least three major groups: ribosomal related genes in the early stage (0-10 hrs), secreted proteins in the intermediate stage( 4-24 hrs) and proteases in the late stage (10-48 hrs). This result provided rich information about the biological changes in the fungal cells after exposure to the nematode (and during starvation). They performed extensive follow-up studies to expand and validate their RNA-seq results. Using rapamycin treatment, they validated the importance of protein synthesis for trap development. Using mutants defective in protein secretion, they confirmed the importance of the secretome to trapping the prey. Using protease inhibitor cocktails and following the invasive growth of the fungus, they confirmed the role of proteases in the late stage of the predation. Overall, the study enriched our understanding about this unique trait and may have implications for the evolution of carnivorous behaviors in plants and fungi in general.

My major critique and suggestion focus on the section describing the evolutionary features of the DUF3129 gene family. I find the question of how the DUF3129 domain containing genes evolved and their potential relationship with the evolution of the unique trapping trait fascinating. However, I think the methods and results are not well described for me to be convinced of their conclusions, i.e., a clade-sepcific expansion and rapid diversification of the "family" contributed to the phenotypic evolution. First, the methods description for this section is very brief. Only a reference for the "orthogroup" analysis was provided, with no specifics on how the analysis was performed, the parameters used, and why the tool is appropriate for their question. The authors stated that they searched 20 "complete" genomes in the main text, while in Figure 4, they showed an analysis that involved 485 fungal genomes. I cannot find information on the species and genome assembly for these species. Since orthology mapping results depend highly on the quality of the assembly and completeness of the genome annotation, I cannot assess whether their conclusion on the uniquely large number of genes containing the DUF domain is reliable. Also, Figure 4B is an orthology map, but didn't reveal the species relationship, making it hard to assess whether the family may have expanded before the NFT clade and lost in other species, or whether it expanded specifically in the former. My suggestions for properly supporting their evolutionary statements are:

1. Carefully choose a set of species that span the Ascomycotetes, with priority given to genomes with comparably high quality of assembly and annotation while considering their life traits relevant for this study (so, ok to sample pathogens, but also need to include related non-pathogens).

2. Explain in detail how the orthogroup analysis was performed, with potential caveats. For a group of key genomes, say the 20 "complete genomes", provide the thresholds used to call hits and the hit list, along with the genome assembly ID information for independent validation.

3. Complement the above analysis with BLAST or HMMER searches of all the genomes to assess whether the gene number results can be independently verified.

4. For the "expansion" claim, they need to reconstruct the evolutionary history of this gene family (the definition of the family also needs to be defined, depending on if all DUF3129 containing genes have that domain alone or whether there are more complex domain architectures, which could complicate the evolutionary analysis). Using gene tree and species tree reconciliation, they can then infer the timing of the gene duplications and losses, in order to distinguish between early-duplication-followed-by-loss vs lineage-specific duplication hypotheses.

5. For the "rapid diversification" claim, I'm not sure what the criteria is for calling this rapid. Is this compared to other genes with similar characteristics? Or just a subjective assessment? Figure S2 contained an alignment that showed relatively poorly aligned regions towards the C-terminus. Without knowing the time of divergence between the proteins aligned, or specific tests for selection, I would conclude that the C-terminus regions are less functionally constrained and thus diverged more rapidly. More generally, sequence divergence alone is not necessarily a "surprise". Follow up analyses or experiments to determine the impact of the variable sequences are needed to properly interpret that finding.

My other question has to do with the secretome analysis. First, I wonder how the authors define "secretome". We know that not all proteins with a signal peptide are secreted outside the cell. Cell wall proteins have the SP but also another signal often at the C-terminus of the protein, to allow it to be inserted into the plasma membrane and later linked to the cell wall. Does the author's definition include both types? I think distinguishing those that are secreted outside the cell and those that are possibly retained on the cell wall has important implications for how they likely contribute to the trapping of the nematode. For example, the authors suggested that the TEP proteins could mediate adhesion through phase-phase separation. How this proposed mechanism would work would depend on whether the protein is anchored on the cell wall or secreted to the vicinity of the fungal cells. Performing a GPI-anchor prediction and separately analyze the secretome (if the definition includes both secreted and cell wall proteins) would help clarify and enrich their results. Second, I'm not familiar with the term "effector" in this particular context, and therefore cannot interpret the "effectome" result. I'd like to suggest that the authors define this term and explain how "EffectorP" works (briefly) to predict them.

Reviewer #2:

In this study, Lin and colleagues examine the transcriptional response of a nematode-trapping fungus during the stages of predation. One signature was the ribosome and DNA replication, and the authors show that rapamycin inhibits trap formation, but rapamycin is a general growth inhibitor and has pleiotropic effects on the cell. Similarly, hydroxyurea is a general growth inhibitor as DNA replication is needed for cellular replication. The signature for the predicted effectors seems much more specific, and the data and conclusions regarding the Sso2 mutant is well supported. 

Additionally, the authors identified multiple secreted proteins, and showed the importance of secreted effectors in trap formation. Excitingly, the authors identified an expanded family of secreted proteins that are induced upon nematode exposure and are enriched in the traps. These proteins are required for trap function and potentially represent a new class of proteins. The gene expression signatures also pointed to a role for secreted proteases, and both chemical protease inhibition and specific deletion of highly induced proteases resulted in decreased nematode digestion. 

Overall, this is a clear, well-written paper with beautiful figures. There are a few minor points that would increase confidence in the interpretation of a few results, but this overall is a beautiful example of how RNAseq can lead to interesting hypothesis that are then followed up with clear and well-controlled mechanistic experiments. 

One limitation of this study is that it relies on GO term annotation to find enriched processes in the RNAseq analysis. However, the total GO term annotation available for A. oligospora is very limited. The best annotated genes are going to be those with clear conservation across ascomycota, but this will ignore all of the organism-specific gene functions and all of the organism-specific genes. There may indeed be a statistically significant enrichment for ribosome biogenesis, but this may also be an artefact of the available data. Based on this, and their chemical data, it would be helpful to show that the dose of inhibitors used is not impacting overall growth.

Reviewer #3:

[identifies himself as Steven D Harrris]

In this manuscript, the authors exploit an Arthrobotrys oligospora-C. elegans pathosystem to provide novel insight into mechanisms that underlie fungal carnivory. Specifically, in response to nutrient limitation, A. oligospora generates adhesive nets that trap nematodes that are then digested. Although there are a growing number of "model systems" utilized to understand the diverse morphologies involved in nematode trapping, the Arthrobotrys system is one of the better developed examples. Here, a well designed and executed global transcriptomic approach provides an unbiased perspective that enables three major findings; 1) in preparation for trap formation, ribosome biogenesis and nuclear division are upregulated, 2) an expanded family of proteins that possess the DUF3129 domain play a key role in trap function, possibly as adhesive factors, and 3) metalloproteases are needed for normal levels of nematode digestion. In addition to these specific results, the study also provides a large compendium of gene expression throughout the trap development process that will serve as a robust resource for future studies. In my view, these novel results are notable because they represent one of the first uses of an unbiased approach to understanding trap formation and function. It is only through the use of these types of approaches that real progress will occur in understanding clade-specific morphologies relevant to specific fungal lifestyles.

I do have the following concerns that should be addressed by the authors;

Lines 111-114. It would be helpful if the authors quantified the different stages of trap development as shown in the images provided in Fig. 1. For example, across each replicate, how many worms were trapped at the 10h timepoints (all, 50%)?

Line 113. What do the authors mean by "worms were washed", and why only at these time points?

Line 276. When comparing the new reference strain to the older one, the authors use the term "strongly". In my view, this is too vague. Even though this is described in more detail elsewhere, some additional context would be helpful here.

Line 335. Given that the fungus is ramping up for trap development under starvation conditions by inducing ribosome biogenesis and nuclear division, it seems reasonable to speculate that autophagy might play a role in scavenging from hyphae that do not form traps. Do the authors detect any signs in their gene expression dates or microscopy analysis that would hint at a role for autophagy?

Lines 327 and 339. Although it seems likely that both TOR and HU operated through their known modes of action to inhibit trap formation, this is not in fact shown by the authors. I would encourage them to provide confirmatory evidence that, for example, HU did block nuclear division as expected.

Figs. 2D, 3E, and 4D. In both cases, it would be helpful if arrowheads were used to denote traps.

Lines 371 and 519. Is there any evidence for genomic clustering of predicted effectors as observed in plant pathogenic fungi? Along the same lines, did the authors observe any patterns in the genomic locations of the amplified DUF3129 family members? For example, are they localized near chromosome ends?

Fig. 3E. Even if described elsewhere, it would be helpful for the reader if the authors noted here what SR2200 is.

Lines 385 and 475. Although complemented controls are provided for the tep1 deletion mutants, they are not for the sso2, mep1, and amp1 deletions. While I see the rational behind this, the inclusion of these controls is typically a key confirmatory step.

Fig. 4G. It is really hard to discern much from these images, even when they are magnified. Arrows and arrowheads might help with denoting key features. Also, in the legend (line 890), the phrase "after 24h of 6h exposure" is not clear.

Lines 440-444 and Fig. 5. These could conceivably be shifted to supplemental files.

Line 463. I would suggest that the authors provide some indication of the effectiveness of the PIC cocktail. The delay in digestion would make sense if the cocktail only partially blocked protease activity. If it caused full inhibition, it would imply that other (perhaps non-proteolytic processes) are involved.

Lines 484-489. The authors should be careful with the use of the term ploidy. Their results document local increases in nuclear numbers. They provide no data on whether the nuclei themselves are haploid, diploid, etc. I would simply state nuclear numbers instead of ploidy.

General comment. Do the authors know whether localized nuclear divisions themselves serve as a cue for subsequent trap formation? For example, in the HU experiment, does the presence of traps (albeit last lower levels than untreated controls) indicate that their formation is uncoupled from nuclear division, or is it simply a technical issue (i.e., ensuring that all regions of hyphae experience similar doses of HU)?

---

## [Decision Letter · Decision Letter 2]

16 Oct 2023

Dear Ping,

Thank you for your patience while we considered your revised manuscript "Dynamic transcriptional profiling in a nematode-trapping fungus reveals key processes required for various stages of fungal carnivory" for publication as a Research Article at PLOS Biology. This revised version of your manuscript has been evaluated by the PLOS Biology editors, the Academic Editor, and two of the original reviewers.

Based on the reviews and on our Academic Editor's assessment of your revision, we are likely to accept this manuscript for publication, provided you satisfactorily address the remaining points raised by reviewer #1. Please also make sure to address the following data and other policy-related requests.

IMPORTANT - please attend to the following:

a) Please change your Title to "Key processes required for the different stages of fungal carnivory by a nematode-trapping fungus" - we think that it is not necessary to lead with the methodological approach.

b) Please attend to the minor remaining requests from reviewer #1.

c) Please address my Data Policy requests below; specifically, we need you to supply the numerical values underlying Figs 1BE, 2ABCDE, 3BDFG, 4ABCH, 5ABCD, S1, possibly S3 (I couldn’t open it), S4, S8ABCD, S9, either as a supplementary data file or as a permanent DOI’d deposition.

d) Please cite the location of the data clearly in all relevant main and supplementary Figure legends, e.g. “The data underlying this Figure can be found in S1 Data” or “The data underlying this Figure can be found in https://doi.org/10.5281/zenodo.XXXXX”

e) As mentioned above, I could not open Fig S3; please check whether this file is corrupted and supply an intact version.

f) Please make any custom code available, either as a supplementary file or as part of your deposition.

We expect to receive your revised manuscript within two weeks. 

*Published Peer Review History*

*Press*

Sincerely,

Roland

Roland Roberts, PhD

Senior Editor,

rroberts@plos.org,

PLOS Biology

DATA POLICY:

Regardless of the method selected, please ensure that you provide the individual numerical values that underlie the summary data displayed in the following figure panels as they are essential for readers to assess your analysis and to reproduce it: Figs 1BE, 2ABCDE, 3BDFG, 4ABCH, 5ABCD, S1, possibly S3, S4, S8ABCD, S9. NOTE: the numerical data provided should include all replicates AND the way in which the plotted mean and errors were derived (it should not present only the mean/average values).

CODE POLICY

Per journal policy, as the code that you have generated is important to support the conclusions of your manuscript, we require that you make it available without restrictions upon publication. Please ensure that the code is sufficiently well documented and reusable, and that your Data Statement in the Editorial Manager submission system accurately describes where your code can be found.

DATA NOT SHOWN?

REVIEWERS' COMMENTS:

Reviewer #1:

I appreciate the authors taking the suggestions seriously and did a thorough revision to address them. I'm overall satisfied with their response and believe that the manuscript is much stronger as a result. In particular, the gene tree and species tree reconciliation analysis revealed a clade specific expansion of the DUF family, significantly enhancing the claim that this family's expansion likely contributed to their unique lifestyle. I have a few further suggestions for the authors to consider:

Proper use of "diverge" and "diversification". In the response to my comments and also in the revised text, the authors seem to use "diverge" and "diversify" not only to refer to sequence divergence, but also to refer to gene duplications and losses. To me, these two terms imply sequence evolution unless a specific explanation is provided beforehand. Thus, their use in the text leads to confusion. For example, on lines 496-497, they stated "these DUF3129 genes diverged significantly during the speciation process within NTF (Fig.S3)", but the evidence in Fig. S3 actually showed gene duplication and loss history, with no information on sequence divergence. I suggest changing the text to "the family of genes containing DUF3129 expanded specifically within the NTF. This association implies that their expansion, followed by sequence divergence, likely contributed to the unique biology of the NTF species.".

Another point of clarification: in the reply to my comments on "rapid diversification", the authors listed low sequence homology and short branch lengths in the species tree as evidence. Low sequence homology, especially when it applies to some parts of an alignment, can be explained by lack of functional and hence evolutionary constraint, and thus doesn't necessarily suggest functional diversification. The argument on short branch lengths of the species tree is difficult to follow, and the method for this figure leaves me wondering what the branch lengths actually represent - amount of molecular changes in the DUF family genes, or is the species tree based on concatenation of single-copy orthologroups?

Otherwise, I found the revision satisfactory and would like to congratulate the authors in making this important contribution!

Reviewer #3:

[identifies himself as Steven D. Harris]

The authors have adequately addressed concerns that were identified in the previous version of this manuscript. In my view, it is now suitable for publication.

---

## [Editor Report · Decision Letter 3]

24 Oct 2023

Dear Ping,

Thank you for the submission of your revised Research Article "Key processes required for the different stages of fungal carnivory by a nematode-trapping fungus" for publication in PLOS Biology. On behalf of my colleagues and the Academic Editor, Aaron Mitchell, I'm pleased to say that we can in principle accept your manuscript for publication, provided you address any remaining formatting and reporting issues. These will be detailed in an email you should receive within 2-3 business days from our colleagues in the journal operations team; no action is required from you until then. Please note that we will not be able to formally accept your manuscript and schedule it for publication until you have completed any requested changes.

Sincerely, 

Roli

Senior Editor

PLOS Biology

rroberts@plos.org